# MULTI-TASK REINFORCEMENT LEARNING WITH SHARED-UNIQUE FEATURES AND TASK-AWARE PRIORITIZED EXPERIENCE REPLAY

## ABSTRACT

Multi-task reinforcement learning (MTRL) has emerged as a challenging problem to reduce the computational cost of reinforcement learning and leverage shared features among tasks to improve the performance of individual tasks. However, a key challenge lies in determining which features should be shared across tasks and how to preserve the unique features that differentiate each task. This challenge often leads to the problem of task performance imbalance, where certain tasks may dominate the learning process while others are neglected. In this paper, we propose a novel approach called shared-unique features along with task-aware prioritized experience replay to improve training stability and leverage shared and unique features effectively. We incorporate a simple yet effective task-specific embeddings to preserve the unique features of each task to mitigate the potential problem of task performance imbalance. Additionally, we introduce task-aware settings to the prioritized experience replay (PER) algorithm to accommodate multi-task training and enhancing training stability. Our approach achieves state-of-the-art average success rates on the Meta-World benchmark, while maintaining stable performance across all tasks, avoiding task performance imbalance issues. The results demonstrate the effectiveness of our method in addressing the challenges of MTRL.

## 1 INTRODUCTION

Humans can perform a fixed set of real-world tasks alone, such as everyday household tasks. Multi-task reinforcement learning (MTRL) is a crucial framework for verifying whether the machine also has this capability. Unlike conventional RL methods have to train separate models for each task, MTRL methods aim to handle a fixed set of tasks via a single model, which brings the following benefits: (1) It alleviates the high cost of data collection and training time in conventional RL. (2) It can further exploit the knowledge synergy of tasks. Because of these benefits, MTRL has recently gained attention from researchers (Tanaka & Yamamura, 2003; Borsa et al., 2016; Haarnoja et al., 2018; Yang et al., 2020; Yu et al., 2020a; Sodhani et al., 2021; Sun et al., 2022). We emphasize that it is necessary and practical to develop MTRL methods.

There are two primary challenges posed by the MTRL problem, including (1) the harmonization of shared and unique features from different tasks and (2) the data sampling strategy during MTRL training. Regarding the first challenge, it is intuitive that some knowledge from similar tasks (e.g., opening a door and opening a window) can be shared and transferred for more efficient training. In the meantime, the unique features of each task should also be effectively extracted and preserved, providing enough information for the machine to know how to perform tasks that are less similar to other tasks. Consequently, the harmonization of shared and unique features is critical. Methods that fail to balance shared and unique features may cause a severe performance imbalance of tasks, i.e., a significant gap in the performance of different tasks, as discussed in (Meng & Zhu, 2023). Besides, for the second challenge, the training data in MTRL come from different tasks (stored in each task's replay buffer). How to sample the stored training data (i.e., transitions) from each task is critical so that the model can consistently improve the overall performance and strengthen the performance of more complex tasks. Poor training data sampling strategies may also lead to the performance imbalance of tasks or slow performance improvement.

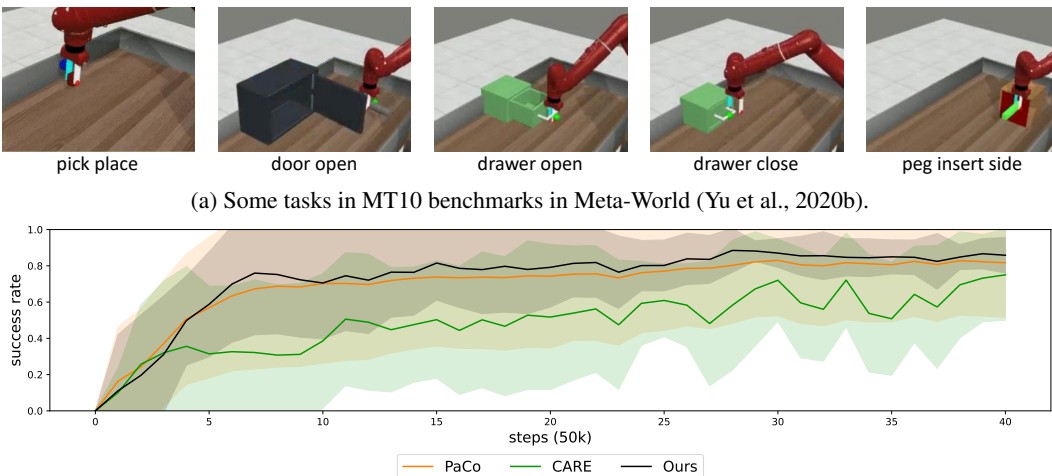

(a) Some tasks in MT10 benchmarks in Meta-World (Yu et al., 2020b).

(b) Showing task performance imbalance problem in MTRL.

Figure 1: Task performance imbalance problem. (a) Some tasks in MT10 benchmarks in Meta-World (Yu et al., 2020b). *door-open* and *drawer-open* may share some related skills while *peg-insert-side* may contains more unique skills. (b) Comparison of average performance using state-of-the-art methods in MTRL on the Meta-World MT-10 benchmark. Results are averaged across different tasks. The larger the standard deviation, the more severe the problem of task performance imbalance. Previous baselinses have encountered serious problem of task performance imbalance. Our solution outperformes the other methods, consistently maintaining a lower standard deviation.

Nonetheless, most existing MTRL methods have not effectively addressed these challenges, resulting in the **task performance imbalance problem**. For example, PaCo (Sun et al., 2022) utilizes a parameter set to extract shared features for all tasks. And Soft Modularization (Yang et al., 2020) and CARE (Sodhani et al., 2021) extract task-specific unique features using a routing network and a pretrained language model-based metadata, respectively. These different approaches neither utilize both shared features and unique features simultaneously, leading to result in Figure 1b. PaCo achieves fast convergence and overall good average performance. However, it exhibits a significant standard deviation across different tasks at each step, indicating that the lack of unique features hinders its effective training on certain tasks, which leads to sub-optimal performance and slower convergence for such tasks. On the other hand, CARE utilizes task-specific unique features, resulting in relatively balanced performance across tasks. However, due to the under-utilization of shared features, it exhibits slower convergence speed as shown in Figure 1b.

Furthermore, current MTRL algorithms mostly use the basic experience replay approach and have not incorporated additional techniques in this aspect. We believe that this can lead to conflicts in the loss functions of different tasks during training, resulting in instability across different tasks in the overall training process, as depicted in Table 1 and Table 6.

In this paper, we propose an approach called shared-unique features along with task-aware prioritized experience replay (PER) to address two challenges in MTRL we mentioned above. Our model compose of two parts: shared features and unique features. We introduce a simple yet effective task-specific embeddings. This approach preserves the unique characteristics of each task, particularly in tasks that require different skills within the task set. By incorporating task-specific embeddings, we can effectively utilize shared skills while preserving the distinctiveness of each task. Additionally, We design the prioritized experience replay (PER) (Schaul et al., 2015) with task-aware modifications to better accommodate multi-task training. With our proposed data sampling strategy, we ensure that the training process gives sufficient attention to all tasks and prevents certain tasks from dominating the learning process. This approach leads to improved stability of the training process across different tasks.

To evaluate the effectiveness of our method, we conduct experiments in the Meta-World (Yu et al., 2020b) benchmark, which provides a diverse set of robotic manipulation tasks, as shown in Fig-

ure 1a. As depicted in Figure 1b, our method achieves state-of-the-art performance across various tasks. Furthermore, our approach consistently maintains a lower standard deviation across different tasks, indicating a more balanced performance and addressing the task performance imbalance problem.

**Key contributions** of this work are *(i)* We address task performance imbalance problem and sample efficiency in MTRL. *(ii)* We propose a simple yet effective task-specific embeddings to preserve the distinct characteristics of each task and help unique tasks' performance. *(iii)* We introduce a task-aware prioritized experience replay (TA-PER) for multi task to enhance training stability across tasks. *(iv)* We achieve state-of-the-art average success rate on the Meta-World (Yu et al., 2020b) benchmark and simultaneously maintain acceptable performance across all tasks.

## 2 RELATED WORK

### 2.1 MULTI-TASK LEARNING

Multi-task learning (MTL) (Caruana, 1997) has been a long-standing problem that has been extensively researched (Andreas et al., 2017; Sener & Koltun, 2018; Zhang & Yang, 2021; Ruder, 2017; Zhang & Yang, 2018; Pinto & Gupta, 2017). In real world, similar tasks are often tackled together, and multi-task learning aims to uncover the relationships among these tasks, utilizing their shared informations to facilitate faster and more effective learning, resulting in a more robust solution. For instance, previous studies (Pinto & Gupta, 2017) have demonstrated that models incorporating a simple parameter sharing structure exhibit superior performance compared to task-specific models in the context of multi-task learning. The main challenges lie in determining which information to share and how to address the **negative influence** of different task losses on a single model.

### 2.2 MULTI-TASK REINFORCEMENT LEARNING

With the recent advancements in reinforcement learning, there has been a growing interest in exploring specialized settings and scenarios, leading to the emergence of multi-task reinforcement learning (MTRL). As a result, several solutions have been proposed to address the challenges associated with MTRL (Tanaka & Yamamura, 2003; Borsa et al., 2016; Haarnoja et al., 2018; Yang et al., 2020; Yu et al., 2020a; Sodhani et al., 2021; Sun et al., 2022; Calandriello et al., 2014; Wilson et al., 2007; Vithayathil Varghese & Mahmoud, 2020; Devin et al., 2017; D'Eramo et al., 2020). One such solution is Soft Modularization (Yang et al., 2020), which introduces a routing network to generate soft combinations for different tasks as routes for the base policy network. This allows for efficient sharing of information among tasks. Another approach is Gradient Surgery (Yu et al., 2020a), which aims to mitigate the negative influence problem in MTRL. This method tackles the issue by projecting a task's gradient onto the normal plane of another task's gradient, effectively resolving conflicts that may arise between tasks. In the context of contextual MTRL, CARE (Sodhani et al., 2021) utilizes metadata to determine the representations for each task. This approach leverages the contextual information to effectively handle the shared and unique features of each task. Also, PaCo (Sun et al., 2022) proposes a novel structure that involves obtaining a network parameter vector for each task through a simple matrix multiplication of a task-agnostic parameter set and task-specific compositional vectors. While these methods primarily focus on addressing the issues of sharing information and negative influence in MTRL, there has been limited exploration on effectively preserving both shared and unique information for each task. As a result, training instability and task performance imbalance problem in MTRL still remains a challenge.

### 2.3 OFF-POLICY REINFORCEMENT LEARNING AND EXPERIENCE REPLAY

Off-policy reinforcement learning is a subfield of reinforcement learning that focuses on learning optimal policies from a dataset of previously collected experiences (Mnih et al., 2013). One of the key advantages of off-policy RL is the ability to separate the data collection policy from the policy being learned. This decoupling allows for more efficient exploration strategies, as the behavior policy can prioritize exploration while the learned policy focuses on exploitation using the experiences stored in the replay buffer. This separation also enables the reuse of existing datasets, making off-policy RL more sample-efficient. Furthermore, in recent years, there have been several advancements in

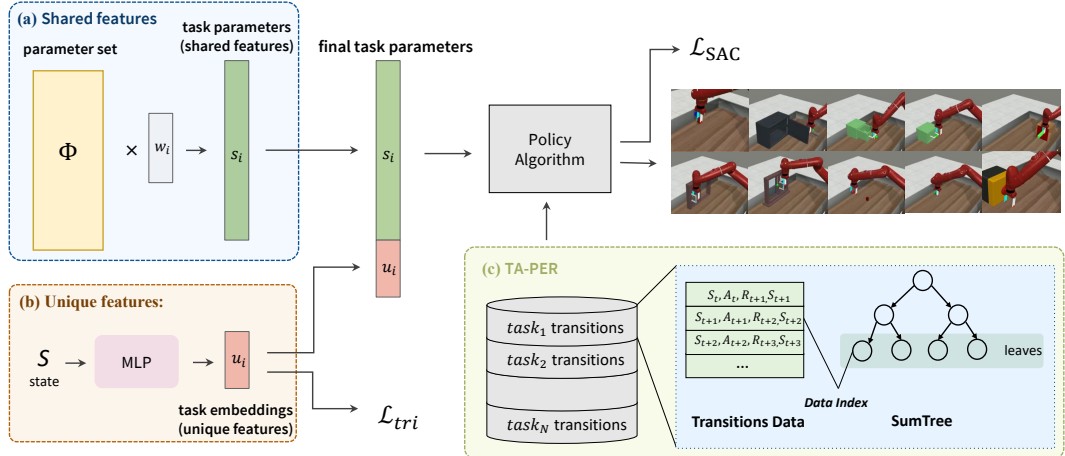

Figure 2: Architecture of our work. Our model compose of (a) Shared features, (b) Unique features, and (c) TA-PER. We update (a) according to $\mathcal{L}_{SAC}$ after apply policy algorithm. We update (b) according to $\mathcal{L}_{tri}$ described in Eqn.(1). In (c), we handle $N$ SumTrees, in each of them we store the priorities in the leaves.

different experience replay methods that further enhance sample efficiency (Wang et al., 2016; Zha et al., 2019; Andrychowicz et al., 2017).

Given these characteristics, many MTRL algorithms utilize the off-policy reinforcement learning framework with experience replay for policy learning, with Soft Actor-Critic (SAC) being one of the most commonly used algorithms (Yu et al., 2020b;a; Yang et al., 2020; Sodhani et al., 2021; Sun et al., 2022). However, as mentioned in section 1, current MTRL algorithms often employ basic experience replay methods without focusing on sample efficiency. This can contribute to the task performance imbalance problem and leaves room for further improvement.

## 3 METHOD

### 3.1 PROBLEM DEFINITION

A **Markov Decision Process** (MDP) is defined by a tuple $(S, A, R, P, \gamma)$, where $S$ represents the state space. $A$ represents the action space. $R : S \times A \to \mathbb{R}$ represents reward function. $P : S \times A \to S$ represents state transition function. $\gamma \in [0, 1)$ represents the discount factors. At each time step $t$, the learning agent get a state $s_t \in S$ and take actions $a_t$ with the policy $\pi(a_t|s_t)$, and move forward to next state $s_{t+1}$ with probability $P(s_{t+1}|s_t, a_t)$ and get a reward $R(a_t|s_t)$. In RL, the agent interacts with an environment and learns to make decisions based on a sequence of states, actions, and rewards. In MTRL, the agent faces multiple MDPs, each corresponding to a specific task, and aims to learn a policy that can maximize accumulated reward of all tasks.

### 3.2 SHARED-UNIQUE FEATURES

We adopted the concept of task-relevant and task-agnostic from (Sun et al., 2022), constructing a parameter set $\Phi = [\phi_1, \phi_2, \ldots, \phi_K]$ as a skill set for tasks $T = [t_1, t_2, \ldots, t_N]$ with vector weight $w_i, i = 1, 2, \ldots, N$ to select the shared skills they require. For each task $i$, **shared features** $s_i$ (referred to as task parameters in (Sun et al., 2022)) is computed as $s_i = [\phi_1, \phi_2, \ldots, \phi_K]w_i$ Furthermore, to address more specific tasks, we developed a straightforward MLP that takes the state as input and generates task embeddings (referred to as **unique features**) $u_i$ specific to each task. These task embeddings are updated using a triplet loss $\mathcal{L}_{tri}$ as:

$$\mathcal{L}_{tri} = max(0, m + d(u_a, u_p) - d(u_a, u_n)) \tag{1}$$

where we sample three unique features $u_a, u_p, u_n$. $u_p$ is same task with $u_a$ and $u_n$ is different task with $r_a$. And function $d$ is the euclidean distance of two features. By utilizing triplet loss, it

---

**Algorithm 1** Training progress of our method

---

**Input:** batchsize K, task number N

 1: **for** each step $n = 1 \ldots MaxEpisodeLength$ **do**
 2:     Calculate $k_i, i = 1, 2, \ldots, N$ refer to **Eq. (5)**
 3:     **for** task $t = 1$ to $N$ **do**
 4:         Sample $k_t$ transitions from $SumTree_t$
 5:         Compute each transition's TD-error $|\delta_j|, j = 1, 2, \ldots, k_t$ refer to **Eq. (2)**
 6:         Update each transition's priority $p_j \leftarrow |\delta_j|, j = 1, 2, \ldots, k_t$
 7:     **end for**
 8:     Calculate shared features $s_i$ and unique features $u_i, i = 1, 2, \ldots, N$
 9:     Concatenate $s_i$ and $u_i$ for following SAC algorithm
10:     Calculate loss $\mathcal{L}_{SAC}$ and $\mathcal{L}_{tri}$
11:     Update parameter set $\Phi$ according to loss $\mathcal{L}_{SAC}$
12:     Update each task's weight according to each task's loss in $\mathcal{L}_{SAC}$
13:     Update MLP according to loss $\mathcal{L}_{tri}$
14: **end for**

---

encouraging a maximized separation between the embeddings of different tasks. Finally, the final task parameter $\theta_i$ is obtained by concatenating vector $s_i$ and $u_i$ and will be utilized for policy training in the SAC algorithm. The complete model architecture is shown in Figure 2. And the whole training and updating progress of the model is shown in Algorithm 1.

### 3.3 TASK-AWARE PRIORITIZED EXPERIENCE REPLAY

We implement prioritized experience replay (PER) (Schaul et al., 2015) and made task-aware modifications to adapt it to SAC. We define the TD-error of a transition sample as the average TD-Error for two Q-networks in SAC:

$$|\delta| = \frac{1}{2} \sum_{i=1}^{2} |r + \gamma V_{\phi_{targ}}(s') - Q_{\theta,i}(s, a)| \qquad (2)$$

Following PER (Schaul et al., 2015), the priority of a transition sample $i$ is defined as $p_i = |\delta| + \epsilon$, where $\epsilon$ is a small positive constant to allow for the sampling of special edge cases where some TD errors are zero. And the probability of transition sample $i$ being sampled is:

$$P(i) = \frac{p_i^\alpha}{\Sigma_{j=0}^{N} p_j^\alpha} \qquad (3)$$

Additionally, when a new transition sample is added in buffer, its priority $p_i$ will be set to maximum priority in buffer, and will update it to $p_i$ after being sampled once. This ensure every transition will be at least sampled once. Furthermore, the importance sampling weight is:

$$w_i = \left(\frac{1}{N} \cdot \frac{1}{P(i)}\right)^\beta \qquad (4)$$

In PER, a data structure called SumTree is utilized to store the priorities in the replay buffer. However, in the context of multi-task settings, if a single SumTree is used in the replay buffer to store priorities of all transitions, it can lead to computational inefficiency and potential issues with some task's transitions not being sampled, resulting in task performance imbalance. Therefore, we construct $N$ SumTree in replay buffer to handle $N$ tasks, each SumTree $i$ handles transition's priority which comes from task $i$. For sampling $K$ transitions from replay buffer, we will sample $k_i$ from SumTree $i$, and $\Sigma_{i=1}^{N} k_i = K$. $k_i$ is computed as:

$$k_i = clip\left(K \cdot \frac{\Sigma_{j=1}^{N_i} p_{ij}}{\Sigma_{l=1}^{N} \Sigma_{j=1}^{N_i} p_{lj}}, k_{min}, k_{max}\right) \qquad (5)$$

where the term $\Sigma_{j=1}^{N_i} p_{ij}$ is the sum of all priorities in SumTree $i$, and the term $\Sigma_{l=1}^{N} \Sigma_{j=1}^{N_i} p_{lj}$ is the sum of all priorities in replay buffer. In addition, we use a clip function to ensure that each task is sampled at least $k_{min}$ times.

## 4 EXPERIMENTAL RESULTS

### 4.1 SETUP

**Benchmarks.** The Meta-World (Yu et al., 2020b) benchmark is a popular evaluation framework for meta-reinforcement learning and MTRL algorithms. It provides a diverse set of robotic manipulation tasks designed to assess the generalization and adaptation capabilities of learning agents in complex and realistic environments. Following (Yang et al., 2020) and (Sun et al., 2022), original multi task benchmarks in Meta-World are fixed goals, so we adopt the MT10-rand setting introduced in (Sun et al., 2022) to better align with real-world scenarios.

**Baselines.** We compare our method with the following baselines: *(i)* Single SAC for each task: Train a separate SAC agent for each task independently. *(ii)* **Multi-Task Soft Actor–Critic (MT-SAC) (Yu et al., 2020b)**: SAC with additional one-hot task encoding as input. *(iii)* **Projecting Conflicting Gradients (PCGrad) (Yu et al., 2020a)**: Project conflicting gradients to mitigate negative influence problem. *(iv)* **Soft Modularization (Yang et al., 2020)**: Propose a routing network that generates routes for different tasks in a modularized manner, allowing the base policy network to adapt to each task. *(v)* **Contextual Attention-based REpresentation (CARE)(Sodhani et al., 2021)**: Utilize task metadata to dynamically decide the representation for each task, allowing for context-aware learning. *(vi)* **Parameter-Compositional Multi-Task Reinforcement Learning (PaCo) (Sun et al., 2022)**: Propose a parameter-compositional structure that separate the task-specific and the task-agnostic parameter set.

### 4.2 RESULTS ON BENCHMARK

The evaluation metric of the multi-task agent is calculated as follows: Every 50,000 steps (for each environment), the agent is tested in the test environment for 50 episodes. The average success rate of each task across these 50 episodes is considered as the performance for that time step. Furthermore, each agent is trained using 10 different random seeds, resulting in 10 success rates for each time step. Finally, the average of these 10 scores is taken as the overall performance for that time step, and the highest mean throughout all time steps is the final evaluation score. The result is displayed in Table 1.

Table 1: Results on MT10-rand (Sun et al., 2022) across 10 different random seeds

| Methods | success rate(%) (mean + std) |
|---|---|
| single SAC for each task | $80.6 \pm 4.2$ |
| MT-SAC (Yu et al., 2020b) | $56.7 \pm 7.5$ |
| PCGrad (Yu et al., 2020a) | $59.4 \pm 8.9$ |
| Soft Modularization (Yang et al., 2020) | $65.8 \pm 4.5$ |
| CARE (Sodhani et al., 2021) | $78.2 \pm 5.8$ |
| PaCo (Sun et al., 2022) | $83.1 \pm 4.6$ |
| Ours | $\mathbf{88.5 \pm 5.3}$ |

Upon observing the table, it is evident that we have exceeded previous state-of-the-art performance. This improvement can be attributed to our successful resolution of the task performance imbalance problem. Specifically, we have taken proactive measures to prevent neglecting or overlooking the training of certain tasks, such as *peg-insert-side*, *push*, *pick-place* as shown in Figure 3, and further discussion will be proposed in 4.3.

Additionally, it is worth mentioning that we follow the discussion in PaCo (Sun et al., 2022) and CARE (Sodhani et al., 2021), incorporating the "single SAC for each task" approach as one of our baselines. This methodology serves as a reference benchmark to showcase the advantages of training individual tasks alongside other tasks, leveraging shared features, within the same number of steps. In our experiments, "single SAC for each task" achieves $80.6 \pm 4.2$ success rate, which is lower than

our performance. This suggests that our method has indeed successfully leveraged the benefits of multi-task training. And the further results on MT50 (Yu et al., 2020b) are provided in Appendix B.

### 4.3 TASK IMBALANCE PERFORMANCE PROBLEM

In multi-task reinforcement learning (MTRL), previous baselines have used different model architectures or employed gradient surgery techniques to mitigate the conflicts in loss functions between different tasks and minimize negative influences. However, due to the inherent uncertainty of reinforcement learning (where results can vary significantly due to random seed initialization), there are still cases where certain tasks exhibit slow convergence or fail to converge altogether, as described in Appendix A.

In our work, we employ TA-PER (Task-aware Prioritized Experience Replay) and additional unique features as introduced in 3.2 and 3.3 to assist in addressing the previously mentioned underperforming tasks. To emphasize the effectiveness of our approach, we have made modifications to the evaluation metric of the multi-task agent, which is outlined as follows:

During training, each agent is trained using 10 different random seeds. Every 50,000 steps (for each environment) in each run, the agent is tested in the test environment for 50 episodes. The average success rate across these 50 episodes is computed, and then the average of these averages across the 10 different random seeds is considered as the performance measure for each individual task at that specific time step. Finally, for every 25,000 steps, the mean and standard deviation of the success rates for the 10 tasks at that time step are calculated. We compare our method with CARE and PaCo, which are two recent state-of-the-art approaches that have achieved high performance in the field of MTRL. The result is displayed in Table 2.

Table 2: Results on MT10-rand at each time step across different tasks.

| Steps per Env (10k) | 25 | 50 | 75 | 100 | 125 | 150 | 175 | 200 |
|---|---|---|---|---|---|---|---|---|
| CARE (Sodhani et al., 2021) | 31.5 ± 37.3 | 39.1 ± 36.8 | 55.0 ± 30.4 | 52.1 ± 39.7 | 60.6 ± 20.2 | 75.8 ± 19.1 | 58.9 ± 28.2 | 78.3 ± 22.9 |
| PaCo (Sun et al., 2022) | 57.0 ± 38.9 | 70.3 ± 44.0 | 73.8 ± 38.1 | 74.3 ± 39.7 | 77.1 ± 32.7 | 83.0 ± 30.6 | 80.6 ± 31.4 | 81.7 ± 30.0 |
| Ours | **58.8 ± 10.8** | **70.5 ± 2.4** | **81.5 ± 6.6** | **79.2 ± 6.7** | **80.2 ± 5.7** | **87.0 ± 3.7** | **85.0 ± 8.4** | **85.8 ± 6.0** |

Table 3: Results of tasks *peg-insert-side* and *pick-place* in MT10-rand at each time step across different seeds.

| Steps (10k) | 25 | 50 | 75 | 100 | 125 | 150 | 175 | 200 |
|---|---|---|---|---|---|---|---|---|
| CARE (Sodhani et al., 2021) | **11.6 ± 5.8** | **19.6 ± 20.1** | **52.0 ± 39.8** | 0.0 ± 0.0 | 60.0 ± 45.0 | 72.4 ± 41.1 | 32.0 ± 42.3 | 35.2 ± 24.9 |
| | 0.0 ± 0.0 | 11.2 ± 0.0 | 16.4 ± 0.0 | 15.6 ± 1.2 | 48.4 ± 0.0 | 35.6 ± 0.0 | 3.6 ± 0.0 | 44.0 ± 0.0 |
| PaCo (Sun et al., 2022) | 0.0 ± 0.0 | 0.0 ± 0.0 | 16.7 ± 37.3 | 16.7 ± 37.3 | 45.7 ± 42.0 | 66.7 ± 47.1 | 64.5 ± 45.8 | 66.2 ± 46.8 |
| | 0.0 ± 0.0 | 0.0 ± 0.0 | 0.0 ± 0.0 | 0.5 ± 1.2 | 0.0 ± 0.0 | 0.0 ± 0.0 | 0.0 ± 0.0 | 0.0 ± 0.0 |
| Ours | 7.0 ± 7.1 | 8.5 ± 8.5 | 33.3 ± 39.2 | **36.4 ± 46.8** | **63.4 ± 45.5** | **78.0 ± 45.3** | **78.4 ± 45.8** | **86.1 ± 44.0** |
| | **49.0 ± 34.1** | **55.2 ± 6.5** | **76.4 ± 20.1** | **72.1 ± 21.8** | **53.8 ± 7.2** | **80.1 ± 16.2** | **78.4 ± 13.7** | **80.1 ± 16.2** |

By designing this modified evaluation metric, we aim to highlight the effectiveness of TA-PER and unique features in assisting under-performing tasks. It allows us to assess the agent's performance across tasks more comprehensively. Moreover, the calculation of the mean and standard deviation for the success rates provides insights into both the average performance and the stability across all tasks of the agent's training process.

For a better understanding of the results, we selected tasks *peg-insert-side* and *pick-place*. In previous works, these two tasks were particularly challenging to train in the MT10 environment (Yu et al., 2020b; Sun et al., 2022). We obtained the results of these two tasks at various time steps for 10 random seeds. The result is displayed in Table 3, demonstrating that our approach effectively assists the agent in maintaining good average performance while aiding in the learning of more difficult tasks.

Table 4: Results of different Sampling Strategies

| Methods | success rate(%) (mean + std) |
|---|---|
| Model with random sampling | $85.1 \pm 4.5$ |
| Model with average sampling | $86.2 \pm 4.3$ |
| Model with PER | $83.3 \pm 5.2$ |
| Model with TA-PER | $88.5 \pm 5.3$ |

## 4.4 SAMPLING STRATEGIES

We have experiments on different sampling strategies to demonstrate the effectiveness of TA-PER. We consider following four methods: *(i)* random sampling *(ii)* average sampling *(iii)* prioritized experience replay (PER) *(iv)* task-aware prioritized experience replay.

Among these four methods, we consider random sampling as the foundational baseline, which has also been the most common approach in previous MTRL practices.

Subsequently, in pursuit of a training process that enables the model to concurrently attend to each task, we experimented with the approach of average sampling. However, in most MTRL libraries, it is already guaranteed that each task has the same amount of new data in the buffer. Due to this factor, the improvement achieved by average sampling in comparison to random sampling is limited.

Following this, we attempted to use prioritized experience replay (PER). However, a situation may arise where certain tasks were neglected and not sampled in a batch because their priorities were overshadowed by the priorities of other tasks, as we mentioned in 3.3. This occurrence was frequent in our experiments, leading to the outcome that using PER resulted in poorer performance compared to random sampling.

Finally, TA-PER calculates an importance weight for each task to determine how many experiences need to be sampled for each individual task. This importance weight is assigned different proportions for each task according to Eqn.(5), and is bounded using $k_{min}$ and $k_{max}$ to prevent situations of excessive or insufficient sampling. By doing so, we were able to adopt the prioritization aspect without encountering the issue of "all the experiences sampled in a batch originate from a single task" we mentioned in PER. Consequently, TA-PER outperformed PER significantly in our experiments.

## 4.5 ABLATION STUDY

In our analysis, we conducted an ablation study on our components to better understand their individual contributions, the result is displayed in Table 5.

Table 5: Results of ablation study.

| Methods | | success rate(%) |
|---|---|---|
| MT-PER | Unique Features | (mean+std) |
| ✗ | ✗ | $83.1 \pm 4.6$ |
| ✓ | ✗ | $86.4 \pm 3.8$ |
| ✗ | ✓ | $85.7 \pm 6.2$ |
| ✓ | ✓ | $88.5 \pm 5.3$ |

**Task-Aware Prioritized Experience Replay.** To demonstrate the effectiveness of TA-PER, we integrated it into several baseline models. The comparison between the baselines and the models with TA-PER is shown in Table 6. By comparing it with Table 1, we can observe that our newly designed experience replay mechanism, TA-PER, indeed improves the learning performance in the multi-task setting. The incorporation of TA-PER helps to prioritize and sample the most relevant and informative experiences for each task, leading to enhanced learning and better overall performance.

Table 6: Results of baselines with TA-PER.

| Methods | success rate(%) (mean + std) |
|---|---|
| MT-SAC (Yu et al., 2020b) | $56.7 \pm 7.5$ |
| MT-SAC (Yu et al., 2020b) + TA-PER | $\mathbf{60.1 \pm 8.3}$ |
| PCGrad (Yu et al., 2020a) | $59.4 \pm 8.9$ |
| PCGrad (Yu et al., 2020a) + TA-PER | $\mathbf{61.3 \pm 10.1}$ |
| Soft Modularization (Yang et al., 2020) | $\mathbf{65.8 \pm 4.5}$ |
| Soft Modularization (Yang et al., 2020) + TA-PER | $60.1 \pm 4.9$ |
| CARE (Sodhani et al., 2021) | $78.2 \pm 5.8$ |
| CARE (Sodhani et al., 2021) + TA-PER | $\mathbf{79.1 \pm 6.9}$ |
| PaCo (Sun et al., 2022) | $83.1 \pm 4.6$ |
| PaCo (Sun et al., 2022) + TA-PER | $\mathbf{86.4 \pm 3.8}$ |

## 5 CONCLUSION

In this paper, by introducing shared-unique features along with task-aware prioritized experience replay, we address the task performance imbalance problem. The utilization of task-specific embeddings preserved the unique characteristics of each task, allowing for more effective learning and improvement for unique tasks' performance. The implementation of TA-PER in the training process facilitated better prioritization and utilization of the collected experiences, resulting in more stable and efficient training across tasks. The experimental results on the Meta-World benchmark demonstrate that our method achieves state-of-the-art average success rates while maintaining stable performance across all tasks, avoiding task performance imbalance issues. The proposed approach shows promise in addressing the challenges of MTRL and improving training stability.

However, there is room for further improvement in our approach. In recent years, various versions of experience replay, such as Hindsight Experience Replay (HER), have been proposed. Integrating these advancements in experience replay into the context of multi-task learning could potentially yield even greater improvements in sample efficiency and training stability.

**Limitations.** As the TA-PER approach utilizes the SumTree data structure, the selection of priorities and updates within the SumTree can lead to higher time complexity compared to the original random sampling approach. This can result in longer overall training times. Exploring ways to reduce the time required for sampling while incorporating the concept of priorities is a potential direction for future research. Furthermore, as the number of tasks increases, the assistance provided by the unique feature set gradually diminishes, as depicted in Appendix B. Incorporating additional metadata and leveraging pretrained language models, as seen in the CARE (Sodhani et al., 2021) approach, holds great potential for further enhancing task-specific embeddings.

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

## A  COMPARISON OF EACH TASK'S PERFORMANCE IN BASELINES

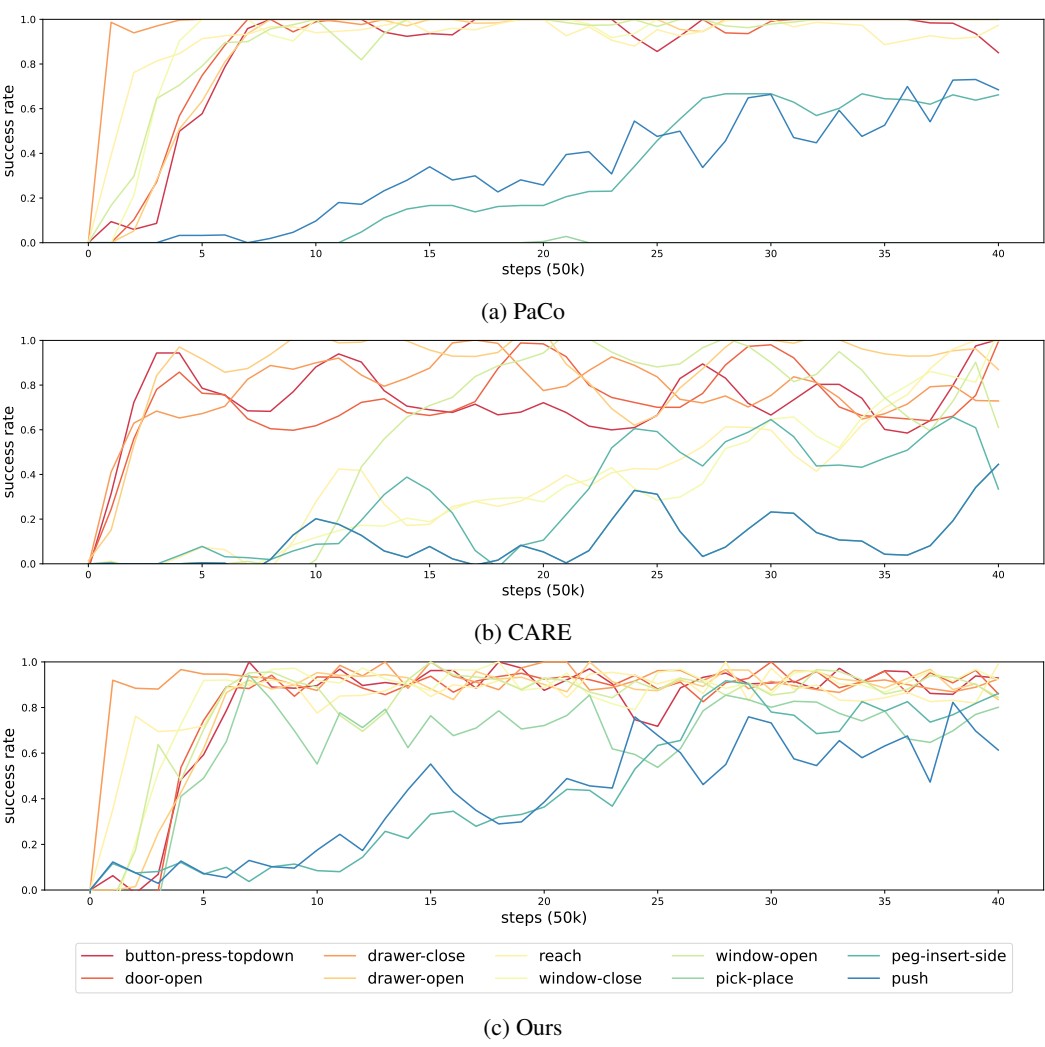

(a) PaCo

(b) CARE

(c) Ours

Figure 3: Comparison of task performance using state-of-the-art methods in MTRL on the Meta-World MT10-rand (Sun et al., 2022) benchmark. Results are averaged over 10 random seeds.

In the comparative analysis shown in Figure 3, we can observe the performance of different methods on specific tasks. Specifically, we analyze the results for PaCo (Figure 3a), CARE (Figure 3b), and our proposed method (Figure 3c).

Table 7: Results on MT50-rand (Sun et al., 2022) across 10 different random seeds

| Methods | success rate(%) (mean + std) |
|---|---|
| MT-SAC (Yu et al., 2020b) | $38 \pm 2.1$ |
| PCGrad (Yu et al., 2020a) | $45 \pm 3.8$ |
| Soft Modularization (Yang et al., 2020) | $48 \pm 2.6$ |
| CARE (Sodhani et al., 2021) | $52.3 \pm 3.7$ |
| PaCo (Sun et al., 2022) | $55.1 \pm 2.4$ |
| Ours | $\mathbf{56.2 \pm 3.1}$ |

For PaCo, as depicted in Figure 3a, it encounters difficulties in tasks such as *peg-insert-side*, *push*, and *pick place*. These tasks exhibit lower performance compared to others, indicating that PaCo struggles to effectively learn and generalize the required skills for these particular tasks.

On the other hand, CARE, as shown in Figure 3b, demonstrates slower performance improvement for tasks like *peg-insert-side*, *push*, and *window-close*. This can be attributed to the under-utilization of shared features in CARE, which hinders its ability to efficiently learn and transfer knowledge across tasks, particularly for tasks that require different and distinct skills.

In contrast, our proposed method, illustrated in Figure 3c, overcomes these limitations and effectively facilitates the learning of tasks that demand specialized and unique skills during training. By incorporating TA-PER and leveraging shared-unique features, our approach enables these tasks to achieve average task performance while also demonstrating remarkable convergence speed. This indicates that our method successfully addresses the task performance imbalance problem and enhances the learning process, leading to improved performance and efficiency across a wide range of tasks.

## B    RESULTS ON MT50

We conducted a comparative evaluation of our proposed method with other baselines on the MT50-rand benchmark (Sun et al., 2022). This benchmark consists of 50 diverse robotic manipulation tasks, and we evaluated the performance using the metrics described in Section 4.2. The results, depicted in the Table 7, were obtained by averaging the performance over 10 random seeds and reporting both the mean and standard deviation.

## C    VISUALIZATION OF UNIQUE FEATURE

Using our simple MLP network and triplet loss, we can bring states belonging to the same task closer together while pushing different task states apart on the MT10 dataset, as evident from the Figure 4. However, as shown in the Figure 5, we did not achieve a clear separation of state embeddings for each task. As the number of tasks increases, we believe that designing more intricate MLP architectures or modifying loss, such as finding a more suitable margin in triplet loss or using other contrastive loss, becomes necessary to prevent issues such as the one depicted in the Figure 5.

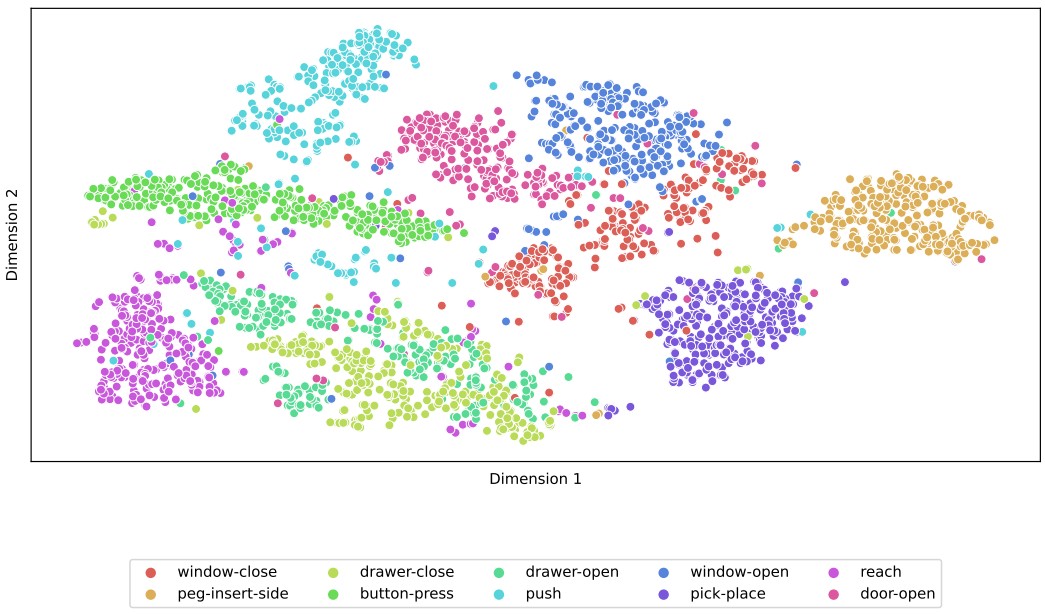

Figure 4: T-SNE visualization of unique features on MT-10.

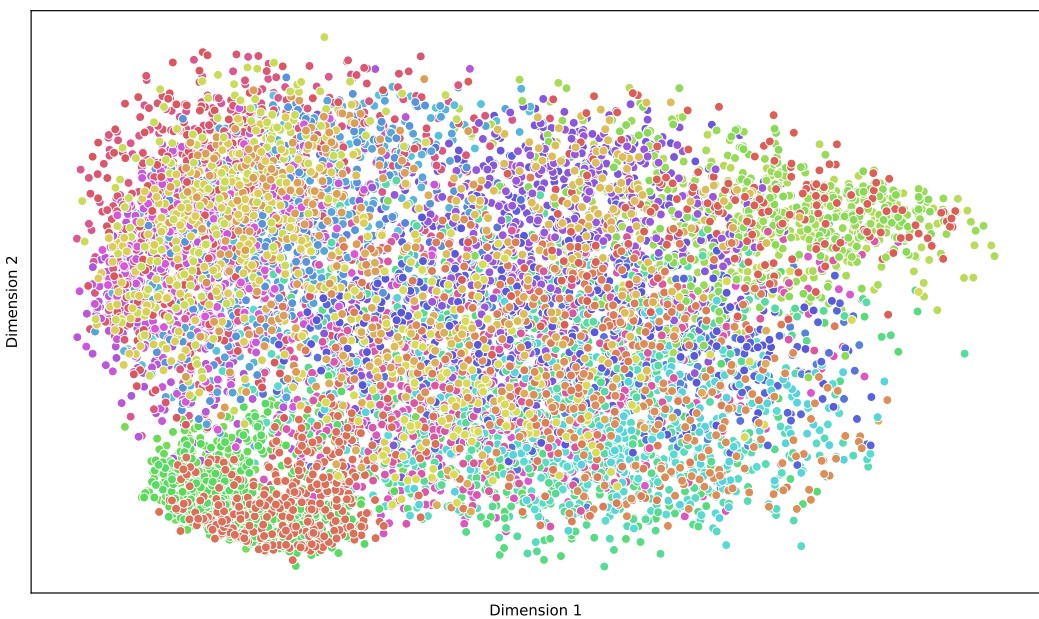

Figure 5: T-SNE visualization of unique features on MT-50.

