# OpenReview forum: "Multi-Task Reinforcement Learning with Shared-Unique Features and Task-Aware Prioritized Experience Replay"
_ICLR.cc/2024/Conference — Submitted to ICLR 2024_

### Official Review · Reviewer_89dx · 2023-10-31

**Soundness:** 3 good
**Presentation:** 3 good
**Contribution:** 2 fair
**Rating:** 3
**Confidence:** 4

**Summary:**

This work discusses Multi-task Reinforcement Learning (MTRL) as a challenging problem aiming to reduce computational costs and improve task performance by leveraging shared features. The key challenge is deciding which features to share across tasks while preserving unique task features to prevent task performance imbalances. The method introduces task-specific embeddings to retain unique task features, addressing potential imbalances of different tasks. Task-aware settings in prioritized experience replay enhance multi-task training stability. The proposed approach achieves state-of-the-art success rates on the Meta-World benchmark, demonstrating effectiveness in handling MTRL challenges.

**Strengths:**

The proposed approach introduces a shared representation and task-based prioritized experience replay, providing a simple and reasonable formulation to address task imbalance performance issues in multi-task reinforcement learning.

Experimental results demonstrate that the proposed method, TA-PER, achieves a notable improvement on MT10 tasks of Meta-World, a widely recognized benchmark for multi-task reinforcement learning. This emphasizes the practical effectiveness of the proposed approach.

The method's simplicity is highlighted as a strength, and the performance improvement is consistently noted throughout the review. This simplicity contributes to the approach's ease of understanding and implementation.

The ablation study further supports the effectiveness of both the shared representation and task-based prioritized experience replay methods on MT10 tasks.

**Weaknesses:**

The review suggests a need for clarification on how unique features are learned. To enhance the understanding, it is recommended to include visualizations comparing representations learned with and without the unique feature loss. This would provide insights into the impact and effectiveness of the unique feature learning process.

The review suggests expanding the comparison with related methods, such as CARE, by including results on MT50 tasks. Providing these additional results would offer a more comprehensive understanding of the proposed method's effectiveness in comparison to existing approaches.

The paper's motivation seems insufficient for acceptance at the ICLR conference. The modeling perspective lacks notable novelty, and the proposed improvements appear incremental rather than transformative. It's essential to note that research, in contrast to engineering, should ideally push the boundaries of existing knowledge and introduce substantial advancements, which may require further refinement to align with the conference's standards.  Overall, the novelty of this work is limited while the performance looks good. Its contribution seems somewhat incremental.

**Questions:**

In addition to the Weakness,

In Figure 1 (b), in steps 5-10, the black line moves out of the black zone. Why does that happen?

Does the unique representation help with the generalization of tasks?

---

### Official Review · Reviewer_MxpY · 2023-11-01

**Soundness:** 2 fair
**Presentation:** 1 poor
**Contribution:** 2 fair
**Rating:** 3
**Confidence:** 4

**Summary:**

The paper presents an approach for multi-task reinforcement learning (MTRL): instead of solving a single MDP, one has to solve a bunch of MDPs from a task family using a single policy. The proposed framework builds heavily on the shared-parameter approach proposed in [1], whereby a policy subspace represented by a set of task agnostic parameters is learned. Policies for all individual tasks lie in this subspace and can be composed by interpolating with the learned set using a task specific weight vector. This paper adds two modifications: 1) use an additional task specific embedding to parameterize the policy, and 2) improve the underlying off-policy RL algorithm using a sophisticated replay buffer sampling [2]. Experiments are conducted on the MetaWorld benchmark and achieves good performance and more importantly, balanced performance across all tasks.

[1] Sun, Lingfeng, et al. "PaCo: Parameter-Compositional Multi-Task Reinforcement Learning." NeurIPS 2022.\
[2] Schaul, Tom, et al. "Prioritized experience replay." arXiv 2015.

**Strengths:**

The paper clearly identifies the task performance imbalance problem in current MTRL methods - large deviation in performance across different tasks - by comprehensively benchmarking the existing methods. Furthermore, the paper also presents a very detailed writeup on the related works encompassing multi-task learning, MTRL and off-policy learning.

**Weaknesses:**

1. The writing of the paper is poor. Section 3, which details the proposed method, starts off with a poor overview of the MTRL setting - it describes the MDP basics in detail but devotes a single line to describing MTRL "...  In MTRL, the agent faces multiple MDPs." I had to refer to [1] to properly understand the setting and associated assumptions. The mathematical notation is incorrect: a) the transition probability function defines a distribution over the states, not a state (it is defined in the text as a probability not a deterministic transition function), b) the reward function is incorrectly defined as a policy R(a_t|s_t). The notation s is used both to define a state and the task-agnostic shared features. The paper jumps to the solution strategy in [1], such as shared parameter set, linear combination of features, etc., without any context and proposes modifications, which is very confusing. I basically had to read [1] to understand even the *notation* of this paper.

2. In terms of the method, it seems the contributions can be summarized in two parts: a) add a state-conditioned embedding in addition to the embedding derived via linear combination of shared parameter set from [1] to parameterize the policy, and b) improve the replay buffer sampling using [2] where the original sampling method is modified to sample from each task with some non-uniform weights. For (a)  I am not sure why the extra embedding is needed since the task ID is provided to the policy as part of the state. I might be wrong here but in my defense no details for the parameter set (such as dimension) is provided as to how the policy is parameterized using these. On the other hand (b) seems like using a better underlying RL algorithm to boost the performance, and not something highly specific to MTRL.

3. No experimental details such as hyperparameters, architecture etc. are provided.

**Questions:**

1. How is the policy parameterized?
2. With the extra embedding/parameter do you use a larger policy compared to using only the shared parameter space policy?
3. What is average sampling?

---

### Official Review · Reviewer_7qsQ · 2023-11-01

**Soundness:** 2 fair
**Presentation:** 2 fair
**Contribution:** 2 fair
**Rating:** 3
**Confidence:** 4

**Summary:**

The authors introduce a multi-task reinforcement learning (MTRL) method to address the performance imbalance across tasks. The proposed method comprises the incorporation of a task-specific embedding into the PaCo approach, along with a task-aware prioritized experience replay (TA-PER). Empirical results show that their proposed method consistently outperforms previous MTRL algorithms across two MTRL benchmarks.

**Strengths:**

The paper proposes an interesting approach to figure out  the performance imbalance across tasks in training through applying the data sampling strategies and and shows empirical improvements on MTRL benchmarks.

**Weaknesses:**

The paper does not introduce a method that is fundamentally different from the approach in prior work[1]. The proposed structure for the policy seems to simply augment the task parameters with task-specific embeddings. Additionally, while the authors claim that the task-aware prioritized experience replay (TA-PER) is a new proposal, it appears to be a minor modification of the existing PER. Although Section 3.2 introduces an approach to extract task-specific features, Figure 5 suggests that the proposed embedding learning method is not effective. The paper also fails to address issues related to complexity and memory requirements. Storing a set of parameters for representation suggests that the proposed method might require significant memory and computational resources. Lastly, the authors should demonstrate the performance on challenging tasks in the MT50 benchmark to establish the efficacy of their proposed method in addressing imbalanced performance. Notably, the paper omits details on the hyperparameters used in the experiments.

Several typographical errors were identified in the paper:

1. In the last line of page 4, the notation should be revised from $r_a$ to $u_a$.
2. In equation (2), the notation of the parameter $\phi_{targ}$ is redundant given the notation of the parameter set.
3. The notation $s_i $ for task parameters is confusing due to its similarity with the notation for state $s$.
4. The notation $w_i$ is redundant, as it can be confused between the importance weight for PER and the vector weight in shared features.

[1] Sun, Lingfeng, et al. "PaCo: Parameter-Compositional Multi-Task Reinforcement Learning." Advances in Neural Information Processing Systems 35 (2022): 21495-21507.

**Questions:**

1.  Please address the concerns in the "Weakness" section and correct the grammar in this sentence

2. In Figure 2, is the unique feature map trained through the both SAC loss and triplet loss?

---

### Meta-Review · Area_Chair_bWy2 · 2023-12-07

**Metareview:**

The proposed method is highly incremental compared with existing methods. Thus, the technical contributions are limited. In addition, experimental results are not comprehensive, and thus not convincing. The presentation also needs to be improved. No authors' responses are submitted.

**Justification For Why Not Higher Score:**

There are many major problems raised by the reviewers.

**Justification For Why Not Lower Score:**

N/A

---

### Decision · Program_Chairs · 2024-01-16

Reject